# Single-Cell Sequencing Analysis Identified ASTN2 as a Migration Biomarker in Adult Glioblastoma

**DOI:** 10.3390/brainsci12111472

**Published:** 2022-10-30

**Authors:** Tangjun Guo, Aijun Bao, Yandong Xie, Jianting Qiu, Haozhe Piao

**Affiliations:** 1Graduate School, Dalian Medical University, Dalian 116000, China; 2Department of Neurosurgery, Affiliated Hospital of Xuzhou Medical University, Xuzhou 221000, China; 3Department of Neurosurgery, Affiliated Hefei Hospital of Anhui Medical University, The Second People’s Hospital of Hefei, Hefei 230000, China; 4Graduate School, Nanjing Medical University, Nanjing 210029, China; 5Department of Neurology, The People’s Hospital of Liaoning Province, Shenyang 110042, China; 6Department of Neurosurgery, Liaoning Cancer Hospital & Institute, Shenyang 110042, China

**Keywords:** single-cell RNA sequencing, glioblastoma, ASTN2, prognosis

## Abstract

Glioblastoma is the most common and aggressive primary central nervous system malignant tumors. With the development of targeted sequencing and proteomic profiling technology, some new tumor types have been established and a series of novel molecular markers have also been identified. The 2021 updated World Health Organization classification of central nervous system tumors first mentioned the classification of adult glioma and pediatric glioma based on the molecular diagnosis. Thus, we used single-cell RNA sequencing analysis to explore the diversity and similarities in the occurrence and development of adult and pediatric types. ASTN2, which primarily encodes astrotactin, has been reported to be dysregulated in various neurodevelopmental disorders. Although some studies have demonstrated that ASTN2 plays an important role in glial-guided neuronal migration, there are no studies about its impact on glioblastoma cell migration. Subsequent single-cell RNA sequencing revealed ASTN2 to be a hub gene of a cell cluster which had a poor effect on clinical prognosis. Eventually, a western blot assay and a wound-healing assay first confirmed that ASTN2 expression in glioblastoma cell lines is higher than that in normal human astrocytes and affects the migration ability of glioblastoma cells, making it a potential therapeutic target.

## 1. Introduction

Glioblastoma (GBM) is the most common and aggressive primary central nervous system (CNS) malignant tumor, with an overall survival of less than 15 months despite standard therapy [1,2]. Conventional treatments, including surgery, radiotherapy and chemotherapy, are limited in their abilities to achieve satisfactory results because of the high invasion and infiltration of tumor cells and chemotherapy resistance. As we know, several risk factors have been identified as established prognostic factors, such as age, histological grade, Karnofsky Performance Status (KPS), extent of resection (EOR) and gene mutation status. In recent years, genetic profiles have been paid increasing attention, and it has been widely accepted that isocitrate dehydrogenase (IDH) mutation, 1p/19q codeletion and O6-methylguanine-DNA methyltransferase (MGMT) gene promoter methylation affect treatment effect and survival [3,4].

With the development of targeted sequencing and proteomic profiling technology, neurooncology researchers have established some new tumor types in clinical practice, and a series of novel molecular markers related to tumor development, treatment and prognosis have also been identified. For this reason, the 2021 updated World Health Organization (WHO) classification of CNS tumors focused on advancing the role of molecular diagnosis in the classification of CNS tumors [5]. At the same time, molecular diagnosis still needs to be combined with established approaches to CNS tumor diagnosis, such as histology and immunohistochemistry. 

In the fifth edition of the WHO classification, one of the most major modifications is the classification of glioma into pediatric and adult types, suggesting that there are clear molecular genetic differences in the occurrence and development of adult glioma and pediatric glioma. Traditional bulk tumor analysis and bioinformatics analysis have identified some key genes, transcriptome changes and pathways that drive malignancy in GBM cells, but they are limited to exploring the diversity and similarity in both types. Instead, single-cell RNA sequencing (scRNA-seq) provides a new approach based on a resolution of an individual cell to reveal the heterogeneity of cancers. Heterogeneity of cancer has a profound impact on the prognosis of patients, and a major aspect of tumor heterogeneity is the tumor microenvironment. Many studies have demonstrated the influence of tumor microenvironment on GBM progression [6,7]. One main application of scRNA-seq is the study of the differentiation and development of tumor cells. Another major application is the determination of the key molecule during the acquisition of malignant potential for tumor cells.

In the present study, we took advantage of scRNA-seq analysis to investigate a detailed dialectical relation between the pediatric and adult types. We also further explored the impact of cellular heterogeneity on adult GBM prognosis and revealed the malignance-associated pathways through which cells gradually acquire the invasive ability. At last, we firstly identified and validated two potential genes involved in adult GBM migration.

## 2. Materials and Methods

### 2.1. Dataset Downloading and Preprocessing

The raw data used in this study were downloaded from the Broad Institute Single-Cell Portal database (https://singlecell.broadinstitute.org/single_cell/study/SCP393/single-cell-rna-seq-of-adult-and-pediatric-glioblastoma/, accessed on 5 August 2022), the Cancer Genome Atlas (TCGA) database (https://portal.gdc.cancer.gov/, accessed on 5 August 2022) and the Gene Expression Omnibus (GEO) database (no. GSE131928), which includes 24,131 cells (7930 cells from Smartseq2 sequencing platform, 16,201 cells from 10X sequencing platform) and 28 samples with GBM. Cyril Neftel et al. [8] accurately divided cells from Smartseq2 sequencing platform into malignant cells and non-malignant cells. As we focused on the analysis of GBM cells, we selected malignant cells from Smartseq2 sequencing platform for analysis, forming a total of 6863 cells and 28 samples (the number of samples was consistent with the original data, without missing any). These cells were further divided into adult and child groups, according to known labels, with 4916 cells and 20 samples in the adult group and 1947 cells and 8 samples in the child group.

### 2.2. Data Normalization and Unsupervised Clustering

The R package “Seurat” is widely used for systematic processing of single-cell sequencing data [9]. Here, we followed the normalization steps. First, genes not expressed in the cells were removed, and a total of 23,686 genes were left. Then, the adult and child group were treated separately. To accelerate the downstream analysis and improve the accuracy, selection of gene features was carried out. The “FindVariableFeatures” function was used to screen the highly variable genes based on the mean.var.plot (MVP) method. A total of 2703 highly variable genes in the adult group and 2989 highly variable genes in the child group were identified.

Next, Principal Component Analysis (PCA) was used for feature extraction of highly variable genes, and JackStraw and Elbow methods were used to select the best number of principal components, which was 54 for the adult group and 25 for the child group. According to the selected principal components, the “FindClusters” function, based on the original Louvain algorithm, was used to perform unsupervised clustering for the two groups, respectively. The clustering results were visualized using Uniform Manifold Approximation and Projection for Dimension Reduction (UMap) methods. The “FindAllMarkers” function was used to identify specific genes for each cell cluster with a significance threshold of adjusted *p* < 0.05 and absolute value of log2FC >1. The similarity between two groups of cell clusters was measured using the Jaccard coefficient. Specifically, the similarity between two cell clusters was equal to the intersection number of specific genes of the two cell clusters divided by the union number of specific genes of the two cell clusters.

### 2.3. Single-Cell Pseudotime Trajectory Reconstruction

To further investigate the similarities and differences in the GBM development process between adults and children from a dynamic perspective, the R package “Monocle” was used to conduct pseudotime trajectory analysis of the two groups of cells. The default DDRTree method of the “reduceDimension” function was used to reduce the dimension, and, subsequently, the “orderCells” function was used to construct the cell pseudotime trajectory. The results of pseudotime trajectories were visualized based on cluster, state and pseudotime, respectively. To reveal the chronological order of each cell cluster in the pseudotime trajectory more clearly, the trajectories were drawn in accordance with cell cluster colors, respectively. The cell states were automatically divided by “Monocle” according to the trajectories, which were mainly based on branches and bifurcation points.

The pseudotime value of each cell was calculated through “Monocle” based on change of gene expression. Each cell was arranged in the trajectory according to the pseudotime value, so as to fit the developmental process of the cell. Therefore, coloring the trajectory based on the pseudotime value can visually reveal the development process of cells. 

### 2.4. Survival Analysis

To investigate the influence of different cell clusters on the prognosis of patients, GBM transcriptome data and corresponding survival data were downloaded from TCGA database. A total of 174 samples were collected, with 59,427 genes, including 169 GBM samples from 160 patients. The CIBERSORT method was used to infer the number of each cell cluster in each sample based on the support vector regression algorithm [10]. The algorithm relies on the known background gene set and scores the fit by gene expression. Since the specific genes reflect the biological characteristics of each cell cluster, we used the specific genes of each cell cluster to construct the background gene set. We divided all the patients into two groups based on the survival time. The patients with longer survival time were assigned to the good prognosis group, and the patients with shorter survival time were assigned to the poor prognosis group. Survival curves were generated using R packages “survival” and “SurvMiner”, and *p* < 0.05 was considered statistically significant using the log-rank test. Box plot was generated using R package “ggplot2” to visually show the differences in each cell cluster between the two groups. Statistical significance was calculated using the Wilcoxon rank-sum test through R package “GGPUBR”.

### 2.5. Prognosis Analysis of Pseudotime Values

To quantify the heterogeneity of different cell clusters, we calculated the pseudotime values for each patient based on the trajectories. The pseudotime value of a cell cluster is equal to the average of the pseudotime values of all cells in the cell cluster; the pseudotime value of a patient is equal to the weighted sum of the pseudotime value of each cell cluster and the proportion of each cell cluster in the patient. Patients were regrouped based on their pseudotime values to evaluate the influence of pseudotime on prognosis using the “surv_cutpoint” function in the R package “SurvMiner”.

### 2.6. Functional Analysis of Key Cell Cluster

Key cell cluster was identified based on trajectory and prognosis analysis. To further evaluate the biological function of key cell cluster, the specific genes of the cluster were identified and visualized by volcano plot. The R package “ClusterProfiler" was used to perform Gene Ontology (GO) functional annotation, Kyoto Encyclopedia of Genes and Genomes (KEGG) pathway enrichment and Gene Set Enrichment Analysis (GSEA) for specific genes. Biological functions or pathways with adjusted *p* values < 0.05 and *q* values < 0.05 were considered significantly enriched and statistically significant. The significantly enriched KEGG pathways were displayed in the form of tree diagrams using the R package “Treemap”, and GO items were displayed in the form of bar graphs.

### 2.7. Identification of Hub Genes

Weighted gene co-expression network analysis (WGCNA) was performed using R package “WGCNA” to explore the key gene modules that regulate cell clusters. Cells and specific genes of the key cell cluster identified previously were used to build the feature expression profiles for WGCNA. To reduce the influence of noisy data, median absolute deviation (MAD) > 0 was used as the threshold to screen genes. Cytoscape software was used for further processing and visualization. We performed the following quality control on nodes and edges in the network: 1. The genes (nodes) assigned to the “Grey” module in WGCNA were deleted because these genes were not assigned to a valid gene module; 2. Since the weight of each edge represents the connection strength between two nodes, the edges with the weight of the top 1% were retained.

### 2.8. Validation of Hub Genes

We verified the validity of hub genes and evaluated the potential association with GBM though gene expression trend analysis, immunohistochemistry (IHC) images and experiments. It is of great significance to analyze the variation trend of a gene in the whole process of cell development from the perspective of pseudotime value and cell state. We drew the scatterplot depicting changes in genes over time using the “plot_genes_IN_pseudotime” function and fitted the change curves. IHC can reflect the expression of genes in tissues. IHC staining images were downloaded from the Human Protein Atlas database for verification.

### 2.9. Cell Culture and Antibodies

The human GBM cell lines, LN229 and A172, were provided by American Type Culture Collection (ATCC, Manassas, VA, USA) and cultured in Dulbecco’s Modified Eagle Medium (DMEM, BasalMedia, Shanghai, China) with 10% fetal bovine serum (FBS, Excell, Suzhou, China). ScienCell provided normal human astrocytes (HAs), which were cultivated in astrocyte growth medium with 5% FBS. All the cells were grown at 37 °C with 5% CO_2_. Abcam provided antibodies against SHISA9 and α-tubulin. Invitrogen provided the antibody against ASTN2. 

### 2.10. Small Interfering RNAs and Transfection 

LN229 and A172 cells were transiently transfected with human ASTN2 small interfering RNA (siRNA); Ribobio (Guangzhou, China) designed and synthesized three siRNAs against ASTN2. The target sequences are listed below:

si3121: 5′-CACCAGTGCTGCTGGAAAT-3′

si3181: 5′-GCACAAAGGAGGCCTTCAA-3′

si2941: 5′-GCAGCAAGAAGGAGCTCAA-3′

All transfections were carried out according to the manufacturer’s instructions using Lipofectamine 3000 (Invitrogen; Thermo Fisher Scientific-CN).

### 2.11. Western Blotting

Total proteins were extracted from cells cultured in vitro and RIPA buffer with protease inhibitors was used to lyse cellular proteins. Equal amounts of protein were added to the electrophoresis tank. Later, the proteins were transferred to polyvinylidene difluoride membrane with 0.45 mm pore size (Millipore, Billerica, MA, USA). The membranes containing the target proteins were tailored according to the Marker and molecular weight of the interest proteins, and then were blocked with 5% milk for 1.5 h at room temperature. The primary antibodies were used to incubate the corresponding target proteins for overnight at 4 °C and then the secondary antibodies were used to incubate the proteins for 2 h at room temperature. At last, the interest proteins were visualized using enhanced chemiluminescence methods.

### 2.12. Wound Healing

Cells grew in 6-well plates for 24 h before being scratched with a sterile pipette tip. Each wound was photographed by inverted microscopy (Leica, Wetzlar, Germany) at 0 and 48 h after rinsing the cells with phosphate-buffered saline (PBS) to remove cellular debris. The total wound area was analyzed using ImageJ software to evaluate the migration capacity.

### 2.13. Statistical Analysis

The results were expressed as Mean ± SEM. Comparisons were performed using Student’s *t*-test with two tails or ANOVA for multiple comparisons. *p*-values less than 0.05 were considered statistically significant. All statistical analyses were performed using GraphPad Prism 9 (Graphpad software inc, San Diego, CA, USA). 

## 3. Results

Our research process is shown in the workflow chart below.

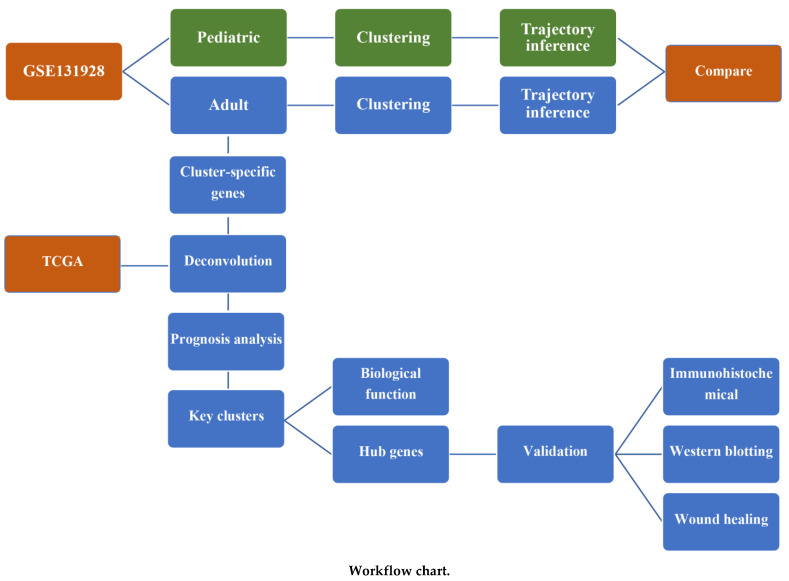


### 3.1. Unsupervised Clustering of Cells Identified Multiple Cell Clusters

All the GBM cells were divided into an adult or child group, with 4916 cells in the adult group and 1947 cells in the child group. After high variable gene screening and dimension reduction by PCA, the adult group was clustered into 15 clusters (Figure 1A) and the child group was clustered into nine clusters (Figure 2A). Considering the effect of cell numbers on cell cluster numbers, we also performed an integrated analysis of adult and pediatric GBM cells. The results also show that the adult GBM cells account for more cell clusters (Appendix A). There were clear boundaries between the clusters, suggesting that these clusters had distinct biological characteristics, which led to the great heterogeneity of the cells. In addition, the number of clusters in the adult group was significantly higher than that in the child group, which indicated that compared with pediatric GBM, adult GBM has a higher level of complexity and heterogeneity and may be more difficult to cure in clinical practice.

To further reveal the biological differences between the clusters, we identified specific genes of each cluster. A total of 14,574 specific genes in the adult group and 14,288 specific genes in the child group were identified. The results show that the expression of specific genes in each cluster was significantly higher than that in other clusters (Figure 1B,C and Figure 2B,C). Most specific genes appeared in both groups. A total of 13,373 specific genes overlapped between both groups, indicating that GBM is evolution-conserved and that only a few genes cause biological differences between children and adults.

### 3.2. Analysis of Development Trajectories Revealed Different Cell Fates

Pseudotime analysis can reproduce the progression of GBM cells. The heterogeneity of GBM can be comprehensively analyzed from a dynamic perspective by linking the process of cell development and cell clusters.

The results show that the adult cell trajectory was divided into 11 time states by five branch points (Figure 3B). In the trajectory, each cell cluster occupied different time sequences (Figure 3A), among which cell clusters 1, 8 and 11 were almost uniformly distributed throughout the development trajectory. Cell clusters 6, 7, 10 and 12 were mainly clustered in the early stage of cell development, while cell clusters 3, 9 and 13 were mainly clustered in the late stage of cell development. The cell trajectory of the child group was divided into five time states by two branch points (Figure 4B). Compared with the adult group, the child group had a simpler pseudotime trajectory. Since there are many factors that affect the comparison of the cell trajectory between adult and pediatric GBM, such as the number of genes and cells we used for the trajectory analysis, whether adult GBM has a higher complexity and heterogeneity than pediatric GBM therefore still needs further investigation. In the child group, cell clusters with a relatively extreme time sequence appeared, such as cell cluster 6, in which almost all the cells were at the end of the trajectory, while those in cell cluster 3 were all in the earliest cell state (Figure 4A).

To assess the potential association between the two groups of GBM cells, the degree of similarity between the adult and child cell clusters was explored based on the Jaccard coefficient. The results show that the cell cluster pairs (adult–child) with strong similarity were 3–6, 5–4 and 3–1 (Figure 5). Combined with the cell development trajectory, it was found that cell cluster 3 in the adult group and cell cluster 1 and 6 in the child group were all advanced cells. However, cluster 5 in the adult group and cluster 4 in the child group were all early cells, which was almost completely consistent with the result of the trajectory inference.

### 3.3. Effect of Cell Heterogeneity on Clinical Prognosis

Significant heterogeneity in tumor tissues may affect the prognosis of patients. Adult GBM has a higher incidence and complexity; therefore, we focused on analyzing the impact of heterogeneity on the prognosis of adult GBM patients.

The 15 cell clusters in the adult group were mapped into GMB tissue data, with clinical survival information downloaded from TCGA by the deconvolution method to construct the sample cell cluster map. According to the survival time, the patients were divided into a good-prognosis group and a poor-prognosis group (Figure 6A). The content differences of the 15 cell clusters between the two groups were presented though paired box plots, and the significance was calculated based on the Wilcoxon rank-sum test. The results show (Figure 6B) that cluster 9 and cluster 10 were significantly different between the two groups (*p* < 0.05). Cluster 9 was more abundant in the poor-prognosis group and cluster 10 was more abundant in the good-prognosis group. 

Combined with the previous cell trajectories, we found that cell cluster 9 was in the middle and late stages of overall cell development, while cell cluster 10 was in the early stage. Therefore, we hypothesized that the state of cell development would have an impact on the prognosis, and that late-state cells would have an adverse impact on the prognosis. To test our hypothesis, the concept of pseudotime value of the patient samples was constructed based on the pseudotime of single cells, and the patients were regrouped into a high-pseudotime-value (late-stage) group and a low-pseudotime-value (early-stage) group. The results show (Figure 7) that there was a significant difference in the survival prognosis between the two groups (Log-rank test *p* = 0.04), suggesting that the time status of cells has a significant impact on the prognosis of patients.

### 3.4. Biological Function Analysis of Cell Cluster 9

Since cell cluster 9 was in the middle and late stage of the trajectory, and its content has a significant impact on the survival prognosis, it was necessary to further analyze the biological function of this cell cluster. A total of 1422 significant specific genes were identified in cell cluster 9. A volcano plot was used to show the differentially expressed genes, among which 238 genes were up-regulated and 1184 genes were down-regulated (Figure 8A). A GO enrichment analysis of these genes showed that they were mainly related to cell adhesion, connection and neuron, axon and synapse development (Figure 8B). A total of 12 biological pathways were significantly enriched by the KEGG enrichment analysis, among which the most significant pathways also included the pathway related to cell adhesion (Figure 8C), which was consistent with the GO enrichment analysis results.

To explore the specific dysregulation direction of the biological processes, a GSEA enrichment analysis was performed. The results show that the most significant up-regulated pathways were mainly related to proteasome and ribosome activities, while the down-regulated pathways were mainly related to antigen-processing and p53 pathways (Figure 8D).

### 3.5. SHISA9 and ASTN2 Are Key Genes Regulating Cell Cluster 9

The biological characteristics of cell cluster 9 are regulated by some key genes. The identification of these key genes provides new ideas for clinical practice. The WGCNA method was used to mine gene modules, and there were three effective gene modules (Figure 9A). These gene modules were then exported in the form of a network and imported into Cytoscape for visualization and further topological property analysis after quality control. In the network (Figure 9B), the degrees of SHISA9 and ASTN2 are significantly higher than those of other genes. We then checked the expression trend of the two genes in the whole development process, and the results show (Figure 10) that the expression trend of the two genes was basically the same. In the early and middle stages, the expression of the two genes showed a slight downward trend and essentially remained stable, while in the late stage, the expression of the two genes increased sharply.

### 3.6. ASTN2 Increases the Migration Ability of GBM Cells

ASTN2 and SHISA9 were identified as the regulator genes of cluster 9 based on the trajectory analysis and WGCNA. The previous KEGG and GO functional enrichment analysis showed that cluster 9 was mainly related to cell adhesion and junction. Cell adhesion ability is closely related to tumor migration, especially the further metastasis of cancer cells. Some studies have demonstrated that the ASTN2 gene played an important role in glial-guided neuronal migration and mutated in neurodevelopmental disorders, including intellectual disability and autism spectrum disorders [11,12]. However, there have been no studies on the relationship between the ASTN2 gene and migration in GBM.

The IHC staining images downloaded from the Human Protein Atlas database showed that ASTN2 is not specific for GBM and that it also expresses moderately in normal glial cells. (Figure 11A). The expression of ASTN2 in GBM tissues downloaded from the TCGA and normal tissues downloaded from the GTEx were examined, and the results show that the expression of ASTN2 in GBM was significantly higher than that in normal brain tissue (Figure 11B). Western blot was used to detect the expression of ASTN2 in the GBM cell lines LN229 and A172 and human astroglia cell line HA. The results show that the expression of ASTN2 in the GBM lines was significantly higher than that in HA (Figure 11C,D). To explore its effect on GBM cell migration, we used the RNA interference to down-regulate ASTN2 expression. After LN229 and A172 cell lines were transiently transfected with ASTN2 siRNAs, we examined the down-regulation efficacy by western blotting. The results show that the ASTN2 expression was reduced significantly by the siRNA compared with the siRNA negative control (siNC) group (Figure 11E,F). Then, we examined the effect of ASTN2 on GBM cell migration by wound-healing assay. We found that control group cells healed the wound to a greater extent than the ASTN2 interference group cells after 48 h (Figure 11G,H and Appendix A).

We also investigated the correlation between ASTN2 and SHISA9 through a bioinformatics analysis, and the result show a significant positive correlation (Figure 12A). Therefore, western blot was used to detect the changes of SHISA9 after the down-regulation of ASTN2. As we predicted, the expression of SHISA9 also decreased after the down-regulation of ASTN2 (Figure 12B,C and Appendix A), which was consistent with the results of our pseudotime trajectory analysis. In summary, these results indicate that the ASTN2 increases the migration ability of GBM cells, and the SHISA9 may be the downstream gene of ASTN2.

## 4. Discussion

In the present study, we took advantage of scRNA-seq analysis to investigate a detailed dialectical relation between pediatric and adult types. The 2021 updated WHO classification of CNS tumors first mentioned the classification of adult glioma and pediatric glioma based on the molecular diagnosis. Here, we used the scRNA-seq analysis to explore the diversity and similarities in the occurrence and development of adult and pediatric types, and to reveal the heterogeneity of cancers. 

The unsupervised clustering of cells identified multiple cell clusters with clear boundaries in adult GBM and pediatric GBM, suggesting that these clusters had distinct biological characteristics and significant heterogeneity. Cell heterogeneity is an important reason for the different prognosis of GBM patients and intertumoral molecular heterogeneity poses a significant challenge for treatment [13]. Other studies classified GBM based on promoter DNA methylation, the microRNA profile and intragenic breakpoints [14,15,16]. Therefore, targeted treatment based on molecular diagnosis is highly necessary and reasonable. Our results also demonstrate that the number of cell clusters in the adult group was significantly higher than that in the child group, which indicates that the adult GBM has a higher level of complexity, heterogeneity and refraction. This was further demonstrated by the pseudotime trajectory analysis. The occurrence and development of tumors is a dynamic process, and cells with different temporal states in the same tissue, are reasons for cell temporal heterogeneity [17]. By reconstructing the cell trajectory and pseudotime based on scRNA-seq data, dynamic processes can be calculated and simulated, which is of great significance for understanding the transition between cell states in cancer [18]. The results show that the adult cell trajectory was divided into 11 time states by five branch points and the child cell trajectory was divided into only five time states by two branch points. Moreover, the distribution of cell clusters in the child group was also relatively simpler than that in the adult group.

However, adult glioma and child glioma are not completely different from each other. Most specific genes of cell clusters appeared in both the adult group and child group, simultaneously. We also found that the cell clusters located in similar time sequences have a higher degree of correlation based on the combination of the Jaccard coefficient and the cell development trajectory analysis. These results indicate that GBM is evolution-conserved between children and adults, and only a few genes cause biological differences. 

The considerable heterogeneity of tumor tissue samples between different patients is an important factor for treatment failure and affects the prognosis of patients [19,20]. We analyzed the differences in adult GBM cell cluster content between different prognostic groups. The results show that cluster 9 was more abundant in the poor-prognosis group and cluster 10 was more abundant in the good-prognosis group. Since cell cluster 9 was in the middle–late stages of the trajectory and cell cluster 10 was in the early stage, we hypothesized that the pseudotime value could be a risk factor of poorer prognosis in adult GBM patients. The survival analysis shows that patients with higher pseudotime values had a worse prognosis. Pseudotime is a measure of the progress of individual cells in processes such as cell differentiation [21]. Therefore, the pseudotime value reflects the state of the cells to some extent, and the effect of pseudotime value on prognosis is actually the effect of cell state on prognosis. Few studies focus on the impact of the pseudotime value on prognosis, and our study found that the pseudotime value is a potential candidate for the impact on prognosis. This may provide new ideas for the establishment of prognostic models in the future.

In 2014, Patel et al. analyzed 430 cells of five primary GBMs though scRNA-seq and found that these cells differed in the expression of various features, including cell proliferation, hypoxic stress, immune response and carcinogenic signaling pathways [22]. Therefore, understanding the functions of single tumor cells and recognizing cell subset characteristics are of great significance for treatment strategies. Identifying specific genes by single-cell gene expression profiling allows the elucidation of mechanisms underlying tumor invasion and migration that are critical for preventing metastasis [23]. Cell cluster 9 had an adverse effect on the prognosis of patients with GBM. The GO, KEGG and GSEA functional enrichment analyses were performed for specific genes of cell cluster 9 to reveal the underlying causes that affect the prognosis. 

The GO enrichment analysis of these genes shows that they were mainly related to cell adhesion, which is closely associated with the development of cancer, especially the further metastasis of cancer cells. The KEGG analysis also reveals that cell adhesion was one of the most significant enrichment pathways, which was consistent with the GO enrichment analysis results. The results of the functional enrichment analysis explain why cell cluster 9 is a poor prognostic factor for GBM patients. Specific cells in cell cluster 9 are involved in the regulation of the cell adhesion function, which promotes the ability of cell invasion and migration, thus leading to poor prognosis of patients. In addition, the migration and metastasis of tumor cells generally occur in the late stage of the disease, which also explains why cell cluster 9 is mainly located in the middle–late stage in the pseudotime trajectory. The GSEA enrichment analysis reveals that the most significant down-regulated pathways were the antigen-processing and p53 pathways. As a tumor suppressor gene, the down-regulation of p53 predicts the further deterioration and development of cancer [24,25], and the weakening of antigen processing ability also leads to the further invasion of cancer cells [26,27]. In conclusion, the malignant characteristics of cell cluster 9 were further confirmed at the level of biological function.

The in-depth analysis of tumor cells at single-cell resolution is conducive to the identification of potential therapeutic targets, contributing to the development of new drugs and the improvement of survival. In 2017, Darmanis et al. performed scRNA-seq on 3,589 cells taken from four GBM patients. A group of genes involved in inhibition of apoptosis, regulation of adhesion and CNS development were identified, which provided new ideas for treatment [28]. In 2019, Wang et al. first identified the RAD51AP1 as an oncogene in GBM using scRNA-seq [29]. This study revealed a new possibility for treatment, which could enhance the therapeutic effect and prolong survival.

The WGCNA method was used based on the scRNA-seq results to investigate the key genes in cell cluster 9, and SHISA9 and ASTN2 were identified as the hub genes. These two genes may play an important role in the process of cell cluster 9 characterizing cancer malignancy. A search in the GeneCards database [30] revealed that SHISA9 may be involved in the regulation of AMPA receptor activity and short-term neuronal synaptic plasticity and was identified as a risk gene for schizophrenia [31]. ASTN2, which primarily encodes astrotactin, has been reported to be dysregulated in various neurodevelopmental disorders [32].

Although some studies have demonstrated that the ASTN2 gene plays an important role in glial-guided neuronal migration, there are no studies about its impact on GBM cell migration. ASTN2 is not specific for GBM and also expresses moderately in normal glial cells. However, both bioinformatics and our experiments show that the ASTN2 expression in GBM was significantly higher than that in normal glial cells. These results suggest that ASTN2 may play a role in the development of GBM. Combined with functional enrichment analysis and its role in glial-guided neuronal migration, it is reasonable to speculate that ASTN2 may have an effect on GBM migration. We found for the first time that the down-regulation of ASTN2 could inhibit the migration of GBM LN229 and A172 cell lines, which provides a new direction for future studies on the inhibition of glioma migration and metastasis. Although the underlying mechanism was not studied further, we found a correlation between ASTN2 and SHISA9. The down-regulation of ASTN2 reduced the expression of SHISA9, which may also be a breakthrough in the study of the mechanism of inhibiting migration.

In summary, our findings demonstrate that pseudotime value maybe a predictive factor for GBM prognosis. It is particularly noteworthy that the overexpression of ASTN2 in GBM cell lines is associated with the expression of SHISA9 in GBM patients. Overall, these findings indicate that ASTN2 could be a promising target for GBM migration inhibition.

## 5. Conclusions

In the present study, we took advantage of scRNA-seq analysis to investigate the relation between pediatric- and adult-type GBM. The number of clusters in the adult group is significantly higher than that in the child group, which indicates that the adult GBM has a higher level of complexity and heterogeneity compared to pediatric GBM. Meanwhile, most specific genes appear in both groups, indicating that GBM is evolution-conserved and that only a few genes cause biological differences between children and adults. These results suggest that adult GBM and pediatric GBM should be treated differently, but not completely separated.

For the first time, we identified the role of ASTN2 in adult GBM migration. This gene shows a higher expression pattern, which is of great importance. This emphasizes the significance of the down-regulation of ASTN2 expression in adult GBM, but the exact mechanism of this gene in these cells remains uncertain, which makes this gene an attractive target for future studies on GBM cells.

## Figures and Tables

**Figure 1 brainsci-12-01472-f001:**
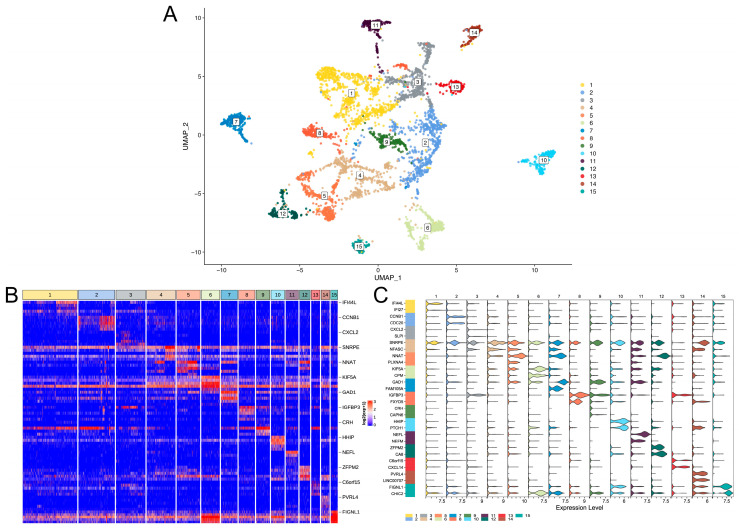
Unsupervised clustering of adult GBM cells. (**A**) UMap dimensionality reduction of clustering results. The X-axis and Y-axis represent the two dimensions of the UMap dimensionality reduction space. The points in the figure are cells, and all cells are colored separately according to the cluster labels. (**B**) The expression heat map of specific genes of cell clusters. Only the top five genes with significant differences in log2FC of each cell cluster are shown, and the gene name is annotated for the first gene. The upper color bar corresponds to each cell cluster. (**C**) Violin diagram of cell cluster-specific gene expression. The upper numeric label and the color bar on the left represent each cell cluster. The X-axis is the expression value, and the Y-axis is the gene. Only the top two genes with significant differences in log2FC of each cell cluster are shown.

**Figure 2 brainsci-12-01472-f002:**
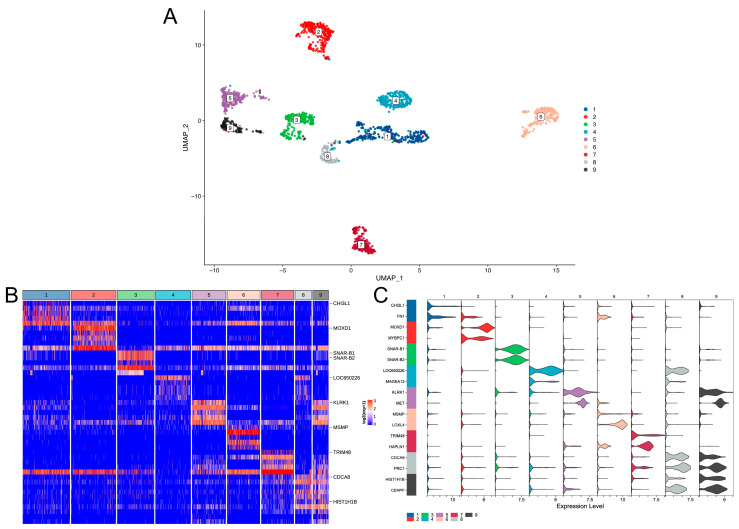
Unsupervised clustering of pediatric GBM cells. (**A**) UMap dimensionality reduction of clustering results. The X-axis and Y-axis represent the two dimensions of the UMap dimensionality reduction space. The points in the figure are cells, and all cells are colored separately according to the cluster labels. (**B**) The expression heat map of specific genes of cell clusters. Only the top five genes with significant differences in log2FC of each cell cluster are shown, and the gene name is annotated for the first gene. The upper color bar corresponds to each cell cluster. (**C**) Violin diagram of cell cluster-specific gene expression. The upper numeric label and the color bar on the left represent each cell cluster. The X-axis is the expression value, and the Y-axis is the gene. Only the top two genes with significant differences in log2FC of each cell cluster are shown.

**Figure 3 brainsci-12-01472-f003:**
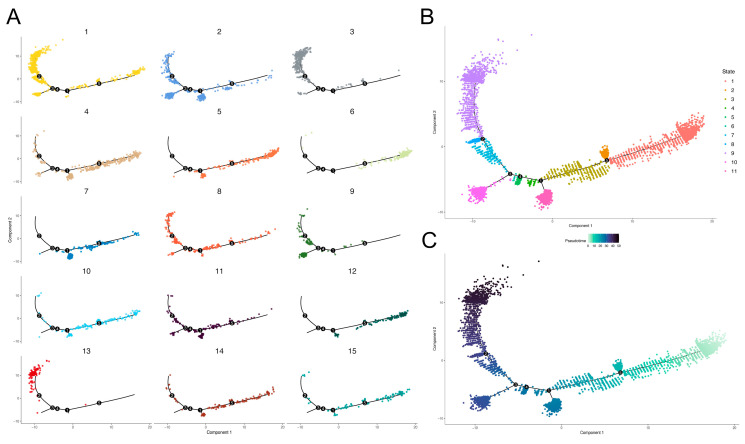
Developmental trajectory of adult GBM cells. The X-axis and Y-axis are two dimensions, the black lines represent the fitted development trajectories and the solid black circles represent branch points. (**A**) Faceted figure of cell clusters. Cell clusters are distinguished by different colors, corresponding to the previous clustering results, and the distribution of each cell cluster in the trajectory is drawn separately. (**B**) Time state of cells. Time states are distinguished by different colors. (**C**) Pseudotime value. The pseudotime value reflects the development direction of trajectory and is represented by asymptotic color, with light color being early stage and dark color being late stage.

**Figure 4 brainsci-12-01472-f004:**
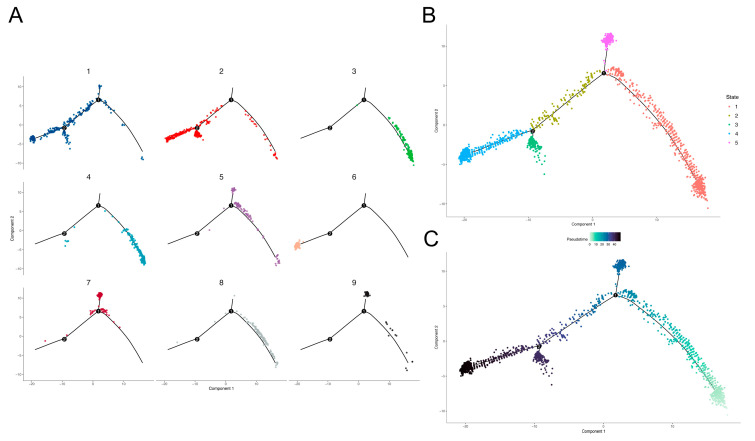
Developmental trajectory of pediatric GBM cells. The X-axis and Y-axis are two dimensions, the black lines represent the fitted development trajectories and the solid black circles represent branch points. (**A**) Faceted figure of cell clusters. Cell clusters are distinguished by different colors, corresponding to the previous clustering results, and the distribution of each cell cluster in the trajectory is drawn separately. (**B**) Time state of cells. Time states are distinguished by different colors. (**C**) Pseudotime value. The pseudotime value reflects the development direction of trajectory and is represented by asymptotic color, with light color being early stage and dark color being late stage.

**Figure 5 brainsci-12-01472-f005:**
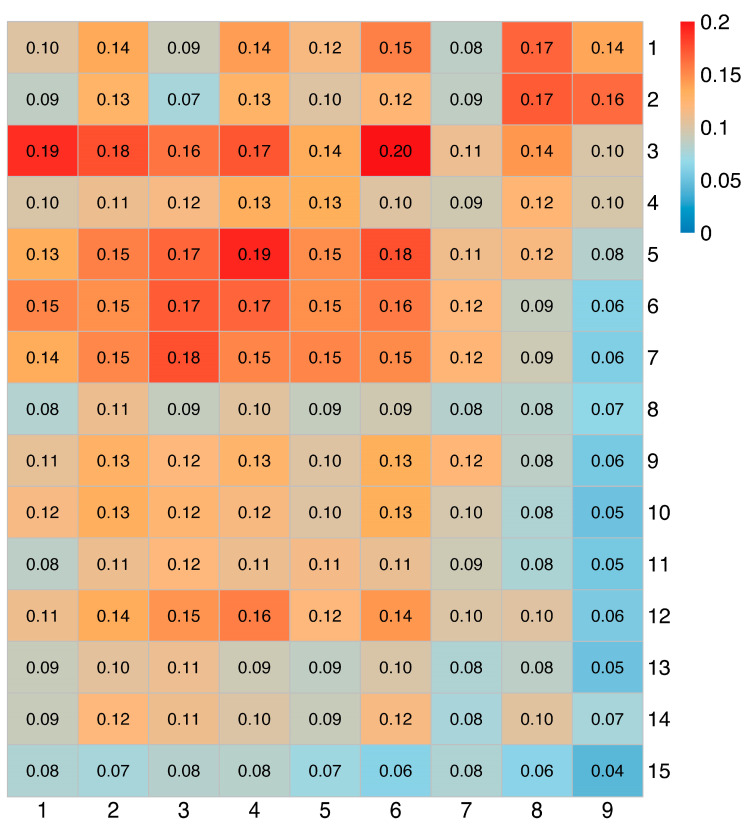
The heat map of similarity of cell clusters between the adult group and the child group. The X-axis is the cell cluster of the child group, and the Y-axis is the cell cluster of the adult group. The values and colors in the figure represent the value of the Jaccard coefficient, namely, the Jaccard similarity.

**Figure 6 brainsci-12-01472-f006:**
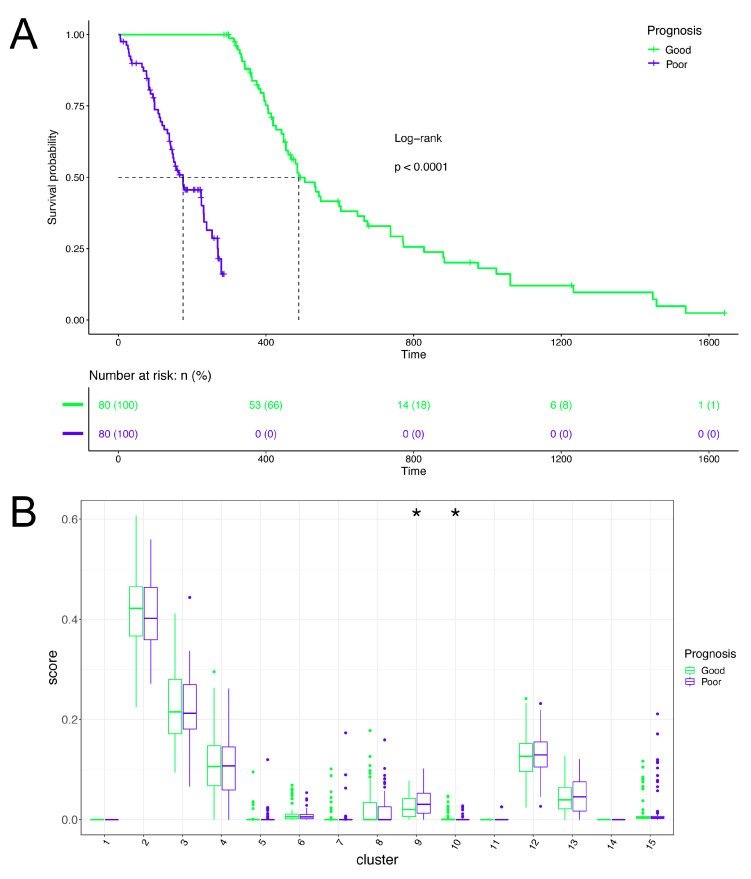
Prognostic analysis of cell clusters. (**A**) Survival curves of different prognosis groups. The X-axis is time (days), and the Y-axis is survival rate. Patients are divided into two groups: good-prognosis and poor-prognosis. Survival curves are drawn, and a log-rank statistical test is performed. (**B**) Box diagram of cell cluster content. The X axis shows 15 cell clusters of adult GBM cells, and the Y axis shows CIBERSORT score (cell content). * represents *p* < 0.05 (Wilcoxon rank-sum test).

**Figure 7 brainsci-12-01472-f007:**
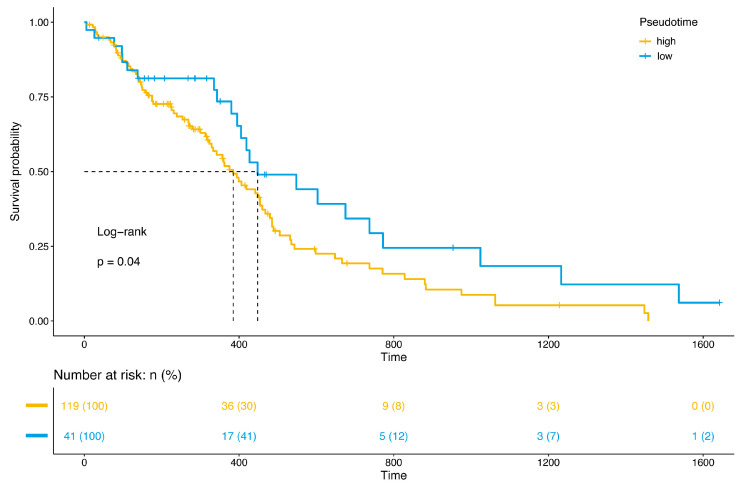
Prognosis analysis of pseudotime. The X-axis is time (days), and the Y-axis is survival rate. Patients are divided into high and low groups based on the pseudotime values. Survival curves are drawn, and a log-rank statistical test is performed.

**Figure 8 brainsci-12-01472-f008:**
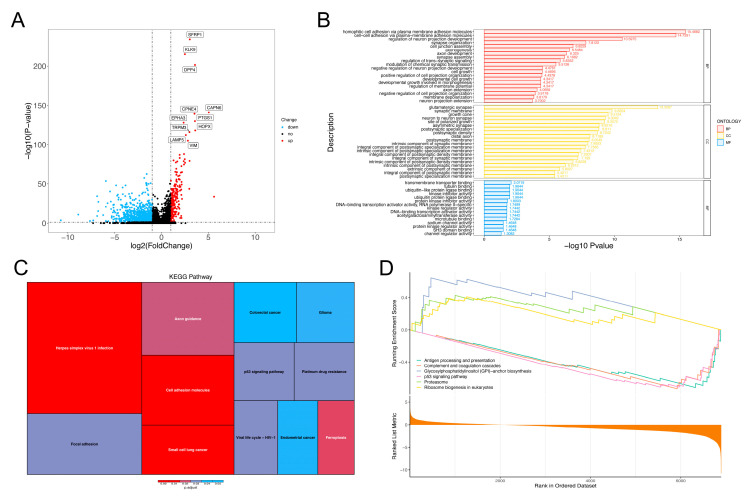
Biological functions of cell cluster 9. (**A**) Volcano map of specific genes. X-axis is log2FC value, Y-axis is -log10 (corrected *p* value). Blue points are down-regulated genes, red points are up-regulated genes and black points are genes with no significant differences. (**B**) GO enrichment bar chart. (**C**) KEGG enrichment analysis. Colors represent the corrected *p* values and areas represent the number of genes enriched into the pathway. (**D**) GSEA enrichment line chart.

**Figure 9 brainsci-12-01472-f009:**
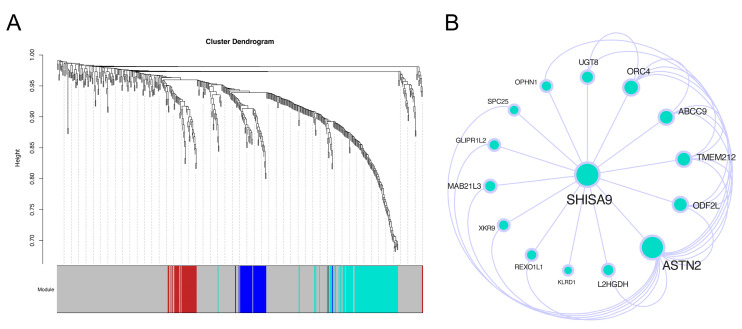
Identification of key genes. (**A**) WGCNA cluster dendrogram. At the bottom are the gene modules, which are colored differently. (**B**) WGCNA network diagram. The node size reflects the degree of the node: the larger the degree, the larger the node size.

**Figure 10 brainsci-12-01472-f010:**
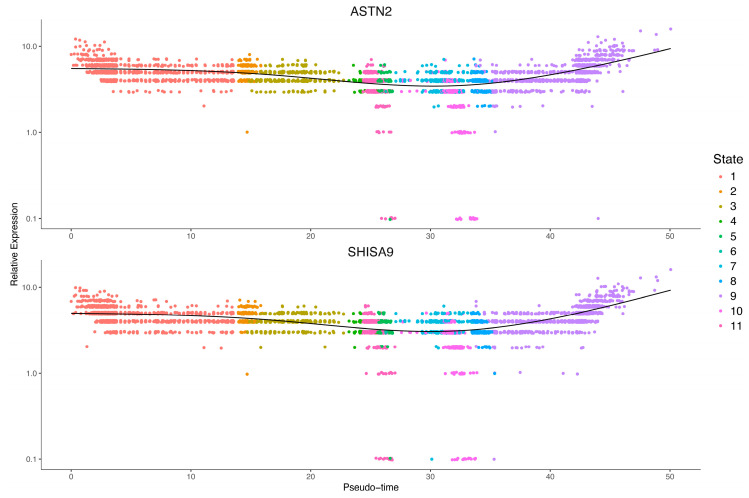
Analysis of gene expression trend. The X-axis is the pseudotime value, and the Y-axis is the expression value. The time states of the cells are shown in different colors.

**Figure 11 brainsci-12-01472-f011:**
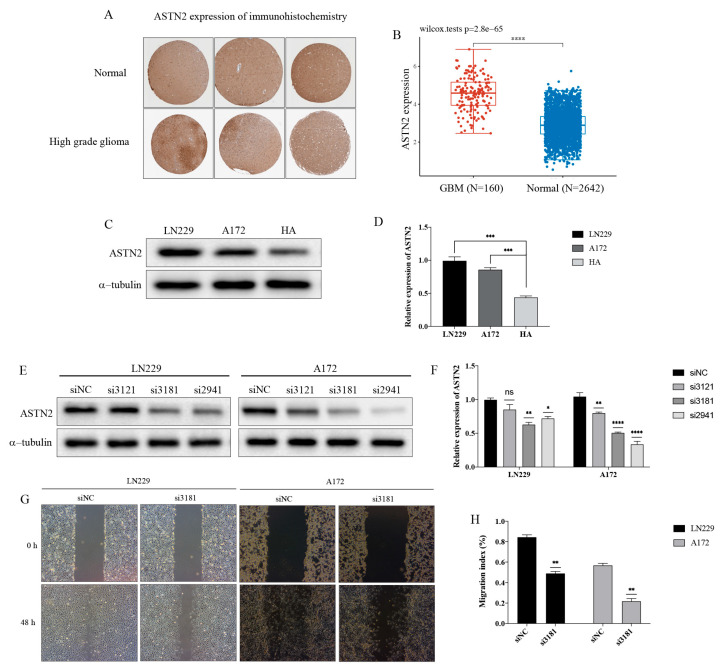
ASTN2 increases the migration ability of GBM cells. (**A**) ASTN2 is not specific for GBM and is also expressed moderately in normal glial cells. (**B**) Bioinformatic analysis shows that the expression of ASTN2 in GBM is significantly higher than that in normal brain tissue. (**C**,**D**) Western blot results show that the expressions of ASTN2 in GBM lines are significantly higher than that in HA. (**E**,**F**) Effects of ASTN2 siRNAs on down-regulation of ASTN2. The results show that si3181 and si2941 are effective in both LN229 and A172 cell lines, and the effect of si3181 is relatively stable. (**G**,**H**) Wound healing assay shows that control group cells healed the wound to a greater extent than the ASTN2 interference group cells after 48 h. **** *p* < 0.0001, *** *p* < 0.001, ** *p* < 0.01, * *p* < 0.05.

**Figure 12 brainsci-12-01472-f012:**
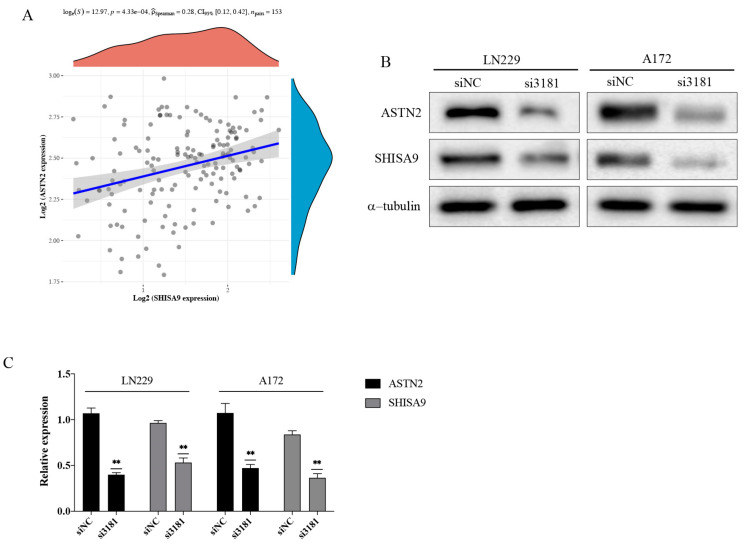
ASTN2 and SHISA9 have a significant positive correlation. (**A**) Bioinformatics analysis results show a significant positive correlation between ASTN2 and SHISA9. (**B**,**C**) Western blot shows that the expression of SHISA9 decreased after down-regulation of ASTN2. ** *p* < 0.01.

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
