# Peer review of "Single-Cell Sequencing Analysis Identified ASTN2 as a Migration Biomarker in Adult Glioblastoma"

_brainsci, 2022, doi:10.3390/brainsci12111472_

Round 1
Reviewer 1 Report
Authors have commendable work using sc RNA sequencing platform to study the difference and similarity in pediatric and adult GBM. Identification and molecular characterization of ASTN2 as a therapeutic target study is well planned.
Author Response
Dear Reviewer,
Thank you very much for your recognition of our study.
For the English language and style, we have commissioned the journal to provide us with English editing services. In our revised manuscript, we have corrected our English grammar errors and typos according to the suggestions provided by the English editing service.
We would like to take this opportunity to thank you for all your time involved and this opportunity for us to improve the manuscript. We hope you will find this revised version satisfactory.
Best regards
The authors

Reviewer 2 Report
In this study, Guo et. investigated the diversity and similarity of the cellular heterogeneity between pediatric and adult glioblastoma and malignance associated maker genes. The authors analyzed the published single-cell sequencing data of glioblastoma and analyzed the different cell clusters, cell trajectory and pseudo-time patterns in pediatric and adult-type glioblastoma. They also investigated the cell clusters in adult-type glioblastoma related to the prognosis of patients and figured out ASTN2 can be a migration biomarker by further validating the discoveries in vitro expression and migration assays using cell lines. In general, the authors discovered some interesting finds in cell heterogeneity differences between pediatric and adult glioblastoma, however, the finds on ASTN2 should be well investigated. There are some points the authors should be addressed.
1.As ASTN2 and SHSA9 are key genes in cell cluster 9, are they related to the prognosis?
2. To avoid the off-targeting effects of ASTN2 siRNAs, please use at least 2 siRNAs for cell migration assays.
3. Please add figure legends and keep the size of each figure consistent.
Author Response
Dear Reviewer,
Thank you very much for your time involved in reviewing the manuscript and your constructive comments on the manuscript.
For the English language and style, we have commissioned the journal to provide us with English editing services. In our revised manuscript, we have corrected our English grammar errors and typos according to the suggestions provided by the English editing serviceWe would like to take this opportunity to thank you for all your time involved and this great opportunity for us to improve the manuscript. We hope you will find this revised version satisfactory.
Please see the attachment.
Sincerely,
The Authors
Reviewer 3 Report
In this study the authors show the diversity and similarity in the occurrence and development in adult and pediatric glioma. Using published single cell RNA sequencing datasets, they rebuilt the cell development trajectory and found aberrant cell fates during glioma development. They also show the correlation between cell heterogeneity and clinical survival. They finally found SHISA2 and ASTN2 are key genes regulating cell cluster 9 and further wet lab validation proved that ASTN2 could increase the migration ability of GBM cells.
Overall the authors try to identify key regulators of glioma development in the adult stage. The study is well designed, however, the following issues should be addressed:
MAJOR CONCERNS
-
Using cell clusters as a parameter to compare cell heterogeneity is not an appropriate way. Since you have a large difference in cell numbers in both the adult and pediatric group, you may not see the same cell culture due to the limit of cell number in the pediatric group. Integration of two datasets is another way to see the difference and similarity between two datasets.
-
Monocle 2 or 3 is a popular way for the cell trajectory analysis, however, it is purely using an informatic algorithm to interpret the cell trajectory. It is still necessary to use RNA velocity to further prove the cell trajectory by RNA splice information.
-
There are many factors that will affect comparison of the cell trajectory between adult and pediatric glioma such as the number of genes you used for the trajectory and the number of cells, it is not appropriate to conclude that the child group has a simpler cell development structure.
-
There is a disconnection between your starting point and clinical prognosis. Does it mean simpler cell heterogeneity has better survival rate in the child group?
Author Response
Dear Reviewer,
Thank you very much for your time involved in reviewing the manuscript and your constructive comments on the manuscript.
For the English language and style, we have commissioned the journal to provide us with English editing services. In our revised manuscript, we have corrected our English grammar errors and typos according to the suggestions provided by the English editing service.
We would like to take this opportunity to thank you for all your time involved and this great opportunity for us to improve the manuscript. We hope you will find this revised version satisfactory.
Please see the attachment.
Sincerely,
The Authors

Round 2
Reviewer 3 Report
The authors have solved my concerns with further exploration. The reviewer has no further question and recommend accepting it.